

# Curcumin inhibits type III secretion of *Pseudomonas aeruginosa*

Miguel Diaz-Guerrero[1], Luis Esaú López-Jácome[2], Rafael Franco-Cendejas[3], Rafael Coria-Jiménez[4], María Guadalupe Martínez-Zavaleta[2], Bertha González-Pedrajo[5], Daniel Huelgas-Méndez[1] and Rodolfo García-Contreras[1]

[1] Faculty of Medicine, Department of Microbiology and Parasitology, Universidad Nacional Autónoma de México, Mexico City, Mexico
[2] Laboratorio de Microbiología Clínica, División de Infectología, Instituto Nacional de Rehabilitación Luis Guillermo Ibarra Ibarra, Mexico City, Mexico
[3] Subdirección de Investigación Biomédica, Instituto Nacional de Rehabilitación Luis Guillermo Ibarra Ibarra, Mexico City, Mexico
[4] Laboratory of Experimental Bacteriology, Instituto Nacional de Pediatria, Mexico City, Mexico
[5] Departamento de Genética Molecular, Instituto de Fisiología Celular, UNAM, Mexico City, Mexico

## ABSTRACT

*Pseudomonas aeruginosa* is a remarkable opportunistic bacterium that represents a global health concern, due to its ubiquity and high levels of antibiotic resistance. Hence, the development of novel antimicrobials or alternative therapies against its infections is urgent. In this regard, anti-virulence therapies are a promising option to minimize pathogen mediated damage to the host rather than directly kill pathogenic bacteria. To date several natural and synthetic compounds had shown activity against quorum sensing regulated virulence factors of *P. aeruginosa*; nevertheless, the type three secretion system (T3SS), also known as injectisome, represents one of the main virulence factors of this bacterium, and a major contributor for acute infections. Importantly, the expression and activity of the injectisome appears not to be positively regulated by quorum sensing, and hence the use of specific quorum quenching enzymes does not inhibit type three secretion. In this work, we characterized the type three secretion profile of effector proteins in a collection of clinical isolates of *P. aeruginosa* isolated from burn patients and respiratory infections. Immunoblotting showed that the presence of an active T3SS is common in these strains, confirming it is an important determinant for its infections. Furthermore, we demonstrate that the natural compound curcumin can effectively inhibit the secretion of the main effectors ExoS and ExoU in PA01 and PA14, the main reference strains of this bacterium, as well as in representative clinical isolates. This inhibition of effectors secretion occurs despite their intracellular accumulation upon curcumin treatment, suggesting that curcumin do not work by impeding effectors expression but rather by interfering with either the assembly or the function of the T3SS.

## INTRODUCTION

Antimicrobial resistance (AMR) is one of the greatest global public health threats facing humanity. In 2019 alone, an estimated 4.95 million deaths were associated with AMR

Corresponding author
Rodolfo García-Contreras, rgarc@bq.unam.mx

pathogens, with 20% occurring in children under five years old. However, the future could be even more dire. The World Health Organization (WHO) predicts that by 2050, AMR-related deaths could rise to 10 million per year. If no action is taken, the World Bank estimates that by 2030, AMR infections could increase global healthcare costs by one trillion USD annually, escalating to 100 trillion USD by 2050 (*WHO, 2023*).

Since 2017, the WHO has published a priority pathogens list highlighting the urgent need for new antibiotics (*WHO, 2024*). The most critical multidrug-resistant bacteria include *Acinetobacter baumannii*, *Pseudomonas aeruginosa*, and various Enterobacterales which are resistant to nearly all antibiotics, including carbapenems third and fourth-generation cephalosporins (*Lopez-Jacome et al., 2019a*; *Lopez-Jacome et al., 2019b*). *P. aeruginosa*, ubiquitous in the environment and metabolically versatile, is a common member of human oral, and intestinal microbiota but acts as an opportunistic pathogen in immunocompromised individuals (*Abram, Jun & Udaondo, 2022*). It possesses diverse virulence factors, including exoproteases, pyocyanin (causing oxidative damage), pyoverdine (siderophore), rhamnolipids, alginate, and toxins delivered *via* protein secretion systems(*Sauvage & Hardouin, 2020*). The type III secretion system (T3SS), or injectisome, is its primary virulence determinant, injecting effectors (ExoS, ExoT, ExoU, ExoY) into host cells (*Tummler, 2022*). ExoS, a 48.3 kDa bifunctional toxin with 453 amino acids, has a GAP domain disrupting actin cytoskeleton *via* GTPases (Rho, Rac, Cdc42) and an ADP-ribosyltransferase (ADPRT) domain targeting Ras proteins and cytoskeletal components, leading to apoptosis (*Horna & Ruiz, 2021*). ExoU, a 73.9 kDa phospholipase with 687 amino acids, destroys host cell membranes *via* its patatin-like phospholipase domain, fully activated by SOD1-mediated ubiquitination and phosphatidylinositol 4,5-bisphosphate (*Sawa et al., 2016*; *Horna & Ruiz, 2021*).

The WHO highlights a critical antimicrobial development crisis, with only 12 new antibiotics approved by the FDA since 2017, most derived from existing drugs with known resistance mechanisms, complicating approval (*WHO, 2022*). To address this, global efforts must prioritize research into alternative strategies, including natural anti-virulence compounds that minimize resistance development. Curcumin, a diarylheptanoid from *Curcuma longa* roots, is a well-studied natural product used for over 4,000 years as a pigment in food and cosmetics (*Abd El-Hack et al., 2021*). Curcumin has shown efficacy against various diseases, including infections, by inhibiting bacterial virulence factors, biofilm formation, and quorum sensing (QS) (*Patel et al., 2020*; *Abd El-Hack et al., 2021*). *Sethupathy et al. (2016)* demonstrated curcumin's ability to reduce QS-dependent virulence factors (*e.g.*, caseinolytic activity, elastase, pyocyanin, and biofilms) in *P. aeruginosa* PA01 and 32 clinical isolates. *Bahari et al. (2017)* confirmed that curcumin impairs QS gene expression (*lasIR*, *rhlIR*), further inhibiting biofilm formation. Although QS and type III secretion system (T3SS) regulation in *P. aeruginosa* are linked, their interaction is complex; some studies suggest RhlR positively regulates *exoS* (*Montelongo-Martínez et al., 2024*), while others indicate QS represses T3SS genes (*Bleves et al., 2005*). Notably, a Δ*lasR*Δ*rhlR* mutant retains ExoS secretion and wild-type infection in a murine model, and lactonase AiiM, which degrades QS signals, does not affect T3SS (*Lopez-Jacome et al., 2019a*; *Soto-Aceves et al., 2019*). These findings suggest that dual inhibition of QS and

T3SS is more effective for anti-virulence therapies than targeting either alone. Our research focuses on natural products like curcumin as alternatives to combat multidrug-resistant bacteria (*Diaz-Nunez, Garcia-Contreras & Castillo-Juarez, 2021*). Hence, in this work, the potential of curcumin to inhibit the T3SS of the two main *P. aeruginosa* reference strains, as well as clinical isolates from burn patients and individuals with cystic fibrosis, was evaluated for the first time, revealing significant T3SS inhibitory activity against most strains.

## MATERIALS & METHODS

### Strains and growth conditions

The reference strains PA01 and PA14, along with clinical isolates of *Pseudomonas aeruginosa*, are part of our laboratory collection. The PA14 Δ*pscD* mutant was obtained from the Ausubel collection (*Liberati et al., 2006*). Clinical strains were isolated from either infected burn patients (P and H strains, from the National Institute of Rehabilitation of Mexico) or pediatric cystic fibrosis patients (INP strains, from the National Pediatric Institute).

Some of the strains were characterized before in terms of their production of QS-dependent virulence factors as parts of previous works (*Garcia-Contreras et al., 2015*; *Campo-Beleno et al., 2022*). Table S1 indicates additional information about the date and source of isolation of each clinical strain.

The study was approved by the Research Committee of the National Institute of Rehabilitation, which determined that patient informed consent was not required for obtaining the clinical isolates (agreement INRLGII 55/22 AC). All strains were grown at 37 °C with constant shaking (250 rpm) in Luria-Bertani (LB) broth or modified LB (10 mM $MgCl_2$, 0.5 mM $CaCl_2$, five mM EGTA) in the experiments in which T3SS was evaluated, with or without 50 µM curcumin dissolved in dimethyl sulfoxide (DMSO; Sigma Aldrich).

Initially the T3SS profile of 22 clinical strains from bacterial collection of Instituto Nacional de Rehabilitación Luis Guillermo Ibarra Ibarra (P and H strains) and from Instituto Nacional de Pediatría (INP strains) was determined. Two reference strains PA01 and PA14 were used as positive control and the PA14 mutant Δ*pscD* was used as a negative control (see Type III mediated secretion assay section).

For further studies, the two reference strains and five clinical isolates were chosen according to their effector's secretion properties: PA01, INP-30R and P044 secrete ExoS but not ExoU and *vice versa*, INP-40, P076 and PA14 secrete only ExoU, while, P193 secreted both ExoS and ExoU.

For those five clinical strains, the minimum inhibitory concentrations of antibiotics (TZP, Piperacillin/Tazobactam; CAZ, Ceftazidime; FEP, Cefepime; ATM, Aztreonam; DOR, Doripenem; IMP, Imipenem; MEM, Meropenem AK, Amikacin; GEN, Gentamicin; CIP, Ciprofloxacin LVX, Levofloxacin CL, Colistin) were determined by the micro broth dilution method in Mueller-Hinton medium following the recommendations of CLSI M07 and breakpoints used defined according (*CLSI, 2022*).

### Effect of curcumin in growth and the production of virulence factors

Once the strains were chosen, we performed growth curves to prove that curcumin does not act as antimicrobial and then could be a selection bias. Each strain was grown in LB overnight at 37 °C and 250 rpm.

For growth curves, we used the LB broth supplemented either with DMSO or curcumin 50 µM dissolved in DMSO. Curves were initiated at a turbidity of $OD_{600\ nm}$ at 0.05 in sterile 96 well plates, incubated at 37 °C with constant 300 rpm orbital shaking, and the increase in turbidity (growth) was recorded every 30 min in a Perkin Elmer Victor Nivo multifunctional plate reader.

Virulence factors were assessed using cultures grown in LB liquid medium for pyoverdine production, caseinolytic activity, and biofilm formation, or in a modified LB medium (supplemented with 10 mM $MgCl_2$, 0.5 mM $CaCl_2$, and 5 mM EGTA) for T3SS-related experiments, as calcium limitation is required to induce T3SS expression *in vitro* (*Horsman, Moore & Lewenza, 2012*), cultures were started at $OD_{600\ nm}$ at 0.05 in two conditions. One group was supplemented with dimethyl sulfoxide (DMSO) that was the solvent used for curcumin (Sigma Aldrich, catalogue number C7727, purity higher than 80%) and the other with 50 µM of curcumin dissolved in DMSO.

### Caseinolytic activity

Overnight cultures of *P. aeruginosa* were used to inoculate (1:200) LB supplemented with either DMSO (curcumin vehicle) or 50 µM curcumin dissolved in DMSO and grown 20 h at 37 °C with shaking. The bacteria were centrifuged for 2 min at 163,300×g and supernatant was collected. Samples of 13 µL of supernatant were incubated with 240 µL of azocasein (1.25% dissolved in 20 mM Tris pH 8.0 and 1 mM $CaCl_2$) at 37 °C for 35 min. Thereafter, 50 µL of mixture was added to 200 µL of $HNO_3$ (1%), gently stirred and centrifuged for 2 min at 16,300×g. Fifty-microliters of supernatant were transferred to 96-well plates and mixed with 150 µL of 0.5 M NaOH and the OD was measured at 405 nm by a Perkin Elmer Victor Nivo multifunctional plate reader (*Loarca et al., 2019*).

### Pyoverdine determination

The supernatant samples collected for the caseinolytic activity measurement were also used for pyoverdine determination by assaying its fluorescence using the method reported in *Garcia-Contreras et al. (2013)* with slight modifications. Three 10-fold serial dilutions for each supernatant were done in 20 mM Tris pH 8.0 and transferred to 96-well black plates, pyoverdine fluorescence was measured with a Perkin Elmer Victor Nivo multifunctional plate reader (excitation 480 nm/emission 560 nm).

### Biofilm formation

Biofilms assay was carried out be described by *Tsukatani, Sakata & Kuroda (2020)* with slight modifications. The turbidity of the bacterial cultures was adjusted to 0.5 $OD_{620\ nm}$. Then, 20 µL of these were used to inoculate 180 µL of LB with or without curcumin 50 µM into wells of flat bottom sterile 96 wells plates. Biofilm was determined independently thrice with five replicas each time. The plate was incubated for 24 h at 37 °C without shaking. After incubation, the used growth media was discarded

and stablished biofilms were washed with distilled water three times and fixed with 200 μL of 100% methanol for 20 min. The solvent was removed and 200 μL of 0.1% violet crystal was added to each well and incubated for 40 min. The plate was then washed three times with distilled water to eliminate the colorant excesses. Then, the plate was dried at room temperature and dyed biofilms were suspended with 200 μL of absolute ethanol. The absorbance was determined at 570 nm in a Perkin Elmer Victor Nivo multifunctional plate reader.

### Type III mediated secretion assay

*P. aeruginosa* overnight cultures were used to inoculate (1:200) modified LB, a medium with limited calcium that allows the expression of T3SS in axenic bacterial cultures (*Dasgupta et al., 2006*). Bacterial cultures were grown in modified LB medium in the absence or presence of 50 μM of curcumin, until an $OD_{600 \text{ nm}}$ 0.8 was reached. One-milliliter of each culture was then centrifuged at $18,000 \times g$ for 2 min, the pellet (containing cells and their intracellular proteins) was frozen until its use and the supernatant was centrifuged once to remove the remaining cells if any, so secreted proteins like ExoS and ExoU became enriched in the supernatants. Secreted proteins from the supernatant were precipitated at 4 °C overnight using 10% v/v trichloroacetic acid (TCA) and centrifuged at $18,000 \times g$ for 30 min at 4 °C to form protein pellets. Both bacterial cell associated and secreted protein pellets were resuspended in Laemmli SDS loading buffer. The supernatant samples were neutralized for residual TCA by adding 10% v/v saturated Tris. All samples were normalized according to the OD of each culture. Proteins were separated on 15% polyacrylamide gels under denaturing conditions and transferred onto nitrocellulose membrane. Proteins were detected by Western blot analysis using anti-ExoU and anti-ExoS polyclonal antibodies and an Immobilon Western chemiluminescent HRP Substrate kit (Millipore). Anti-ExoU and anti-ExoS polyclonal antibodies were raised by subcutaneous immunization of New Zealand rabbits with purified recombinant proteins, as previously described (*Garcia-Ulloa 2nd et al., 2019*; *Soto-Aceves et al., 2019*). ExoS antibodies recognize epitopes present on the ExoT protein. Western blot images were scanned and ExoU and ExoS band intensities were densitometrically quantified using ImageJ software (https://imagej.net/ij/).

### Statistical analysis

Virulence factor determination and T3SS effectors secretion with and without curcumin (analyzed as densitometry values using the Image J software) were determined in triplicate and the mean ± standard error of the mean (SEM) was calculated. The statistical analysis was performed with R (version 2023.06.1; *RStudio Team, 2023*), the normal distribution of the data was verified using a Shapiro–Wilks test and the homoscedasticity using F and Levene tests. Differences were considered significant when the *p*-value < 0.05 in a two-tailed *T*-test.

# RESULTS

In this study, 22 clinical isolates of *P. aeruginosa* were screened for the secretion of protein effectors by the T3SS. Five of them were selected for detailed characterization, INP-30R and

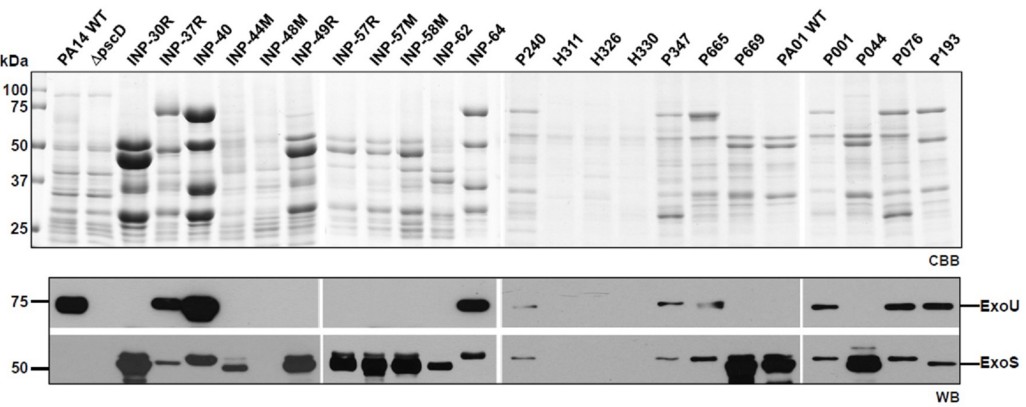

**Figure 1** **Type III Secretion profile of *P. aeruginosa* parental strains PA14, PA01 and clinical isolates solved by SDS-PAGE.** Coomassie brilliant blue stained gel (CBB). Immunoblotting using anti ExoU and anti ExoS polyclonal antibodies (WB).

INP-40 isolates of cystic fibrosis and P044, P076, and P193 from burned patients. These isolates were chosen to have representatives of the isolates from burn patients and from cystic fibrosis with secretion profiles that resemble either the profile of the PA01 reference strain which secretes ExoS but not ExoU (INP-30R and P044) or the profile of the PA14 reference strain which secretes ExoU but not ExoS (INP-40 and P076). In addition, the isolate P193 was also tested since it was able to secrete both ExoS and ExoU.

Regarding their antibiotic resistance, the isolates varied from sensitive (INP-40 and P193), to multidrug resistant (INP-30R and P076) which are resistant to at least three different antibiotics categories to extensive drug resistant (P044) which is resistant to all different antibiotic categories except of colistin (Table S2).

According to the secretion profile of ExoS and ExoU effectors, we defined three groups; the first one which secretes ExoS but no ExoU, including INP-30R, P044 and PA01 strains, the second one secreting ExoU but in which ExoS was absent, it included INP-40, P076 and PA14 strains and the third one including P193 as the only strain in our screening capable of simultaneously secreting ExoS and ExoU (Fig. 1). In the supernatant of the second group, in addition to ExoU, the ExoT effector is also observed due to the polyclonal nature of the serum against ExoS, that have cross reactivity with ExoT since these two proteins share 76% amino acid identity (*Liu et al., 1997*). Interestingly, we identified the secretion of both ExoU and ExoS effectors in the isolate P193, among the 22 screened isolates.

The growth curves of reference and clinical isolates of *P. aeruginosa* were determined over 5 h in LB modified broth supplemented with DMSO (used to dissolve the curcumin) or 50 µM curcumin dissolved in DMSO, and only a slight delay on some bacterial growth rates was observed when curcumin was present in LB cultures (Fig. S1).

To evaluate the impact of curcumin on ExoU and ExoS secretion of *P. aeruginosa* strains, we analyzed the T3SS profiles of wild-type PA14 and PA01, as positive controls, of a PA14Δ*pscD* mutant that is unable to secrete T3SS effectors, as a negative control and the secretion of the selected clinical isolates: INP-30R, INP-40, P044, P076, and P193. Theses

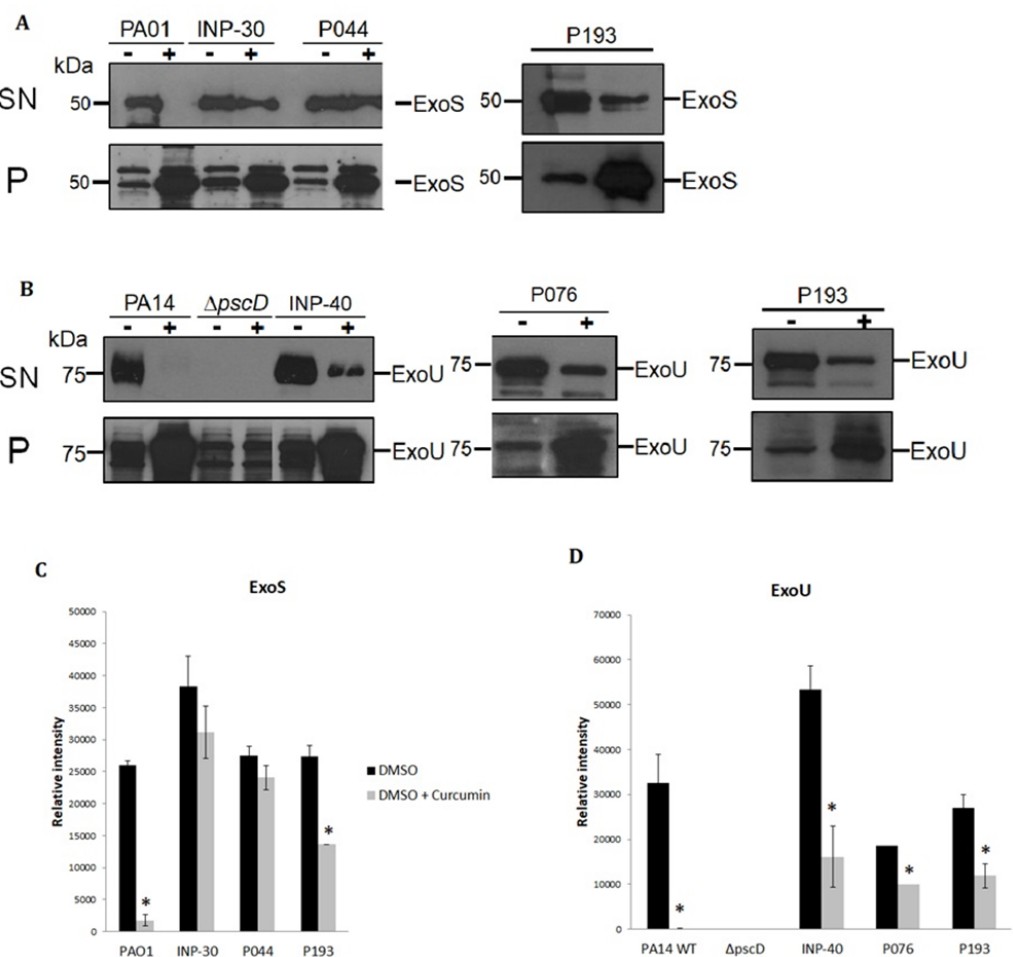

**Figure 2** **Effect of curcumin on effector proteins ExoS and ExoU.** Immunoblotting anti-ExoS (A) and anti-ExoU (B) polyclonal antibodies. Protein samples from supernatant (SN) or bacterial pellet (P) from the indicated strains grown in LB supplemented with (+) or without (-) curcumin 50 uM. Experiments were done in triplicate and one representative replica is shown. Average densitometry values ± SEM of the triplicates evaluating the effect of curcumin in the secretion of ExoS (C) and ExoU (D) are shown; an asterisk (*) indicates that the inhibition shown by curcumin is significant ($P < 0.05$ in a two-tailed $T$-test).

assays were carried out in presence (+) or absence (-) of curcumin 50 µM dissolved in DMSO. We found that curcumin completely inhibited ExoU and ExoS secretion of the parental PA14 and PA01 strains, respectively, and produced a severe impairment of ExoU secretion in INP-40, P076, and P193, and also a strong inhibition of ExoS secretion in P193; whereas, curcumin had only slight (non-significant) inhibition of ExoS secretion in INP-30R and P044 clinical isolates (Fig. 2A). It is important to highlight that this compound increased the ExoU and ExoS protein levels in all bacterial pellets of *P. aeruginosa* evaluated in this work (Fig. 2B). Remarkably, despite the higher production of effectors, the secretion of ExoU and ExoS was either inhibited or unaffected in all tested strains.

Furthermore, we evaluated the QS-regulated virulence factors such as proteolytic activity of extracellular proteases, siderophore pyoverdine and biofilm production by parental
and clinical isolates of *P. aeruginosa* in presence or absence of curcumin. Remarkably, curcumin significantly inhibited the pyoverdine production of all tested strains (Fig. S2A), but had only a slight inhibitory effect on the proteolytic activity of PA14 and PA01 strains, while all clinical isolates exhibited very weak proteolytic activity even when they were cultured without curcumin. Interestingly, P044 exoprotease activity was increased rather than inhibited by curcumin (Fig. S2B). Finally, for biofilm formation curcumin had the tendency to decrease it for all the strains; however, the inhibition was significant only for the clinical isolate INP-30 (Fig. 2C).

## DISCUSSION

The rising threat of AMR underscores the urgent need for innovative therapeutic strategies to combat multidrug-resistant (MDR) pathogens like *P. aeruginosa*. This study provides compelling evidence that curcumin, a natural compound derived from *Curcuma longa*, effectively inhibits the secretion of T3SS effectors, ExoS and ExoU, in both reference strains (PA01 and PA14) and selected clinical isolates in the study. These findings highlight curcumin's potential as an anti-virulence agent, offering a promising alternative to traditional antibiotics, which are increasingly ineffective against MDR strains.

The T3SS is a pivotal virulence factor in *P. aeruginosa*, enabling the direct delivery of toxic effectors into host cells, thereby facilitating acute infections. Our screening of 22 clinical isolates from burn and cystic fibrosis patients confirmed that T3SS activity is prevalent, with distinct secretion profiles for ExoS and ExoU. Notably, the isolate P193, which secretes both effectors, is a rare exception, as mutual incompatibility between *exoS* and *exoU* genes is well-documented (*Kulasekara et al., 2006*; *Horna et al., 2019*). This diversity in effector secretion profiles among clinical isolates underscores the adaptability of *P. aeruginosa* and the importance of targeting the T3SS to mitigate its pathogenicity.

Curcumin's ability to inhibit ExoS and ExoU secretion across reference and clinical strains is a significant finding. Importantly, the increased intracellular accumulation of these effectors in curcumin-treated cells suggests that curcumin does not suppress effector expression but rather disrupts T3SS assembly or function. Previous studies have shown that QS inhibitors, such as lactonase AiiM, fail to affect T3SS activity (*Lopez-Jacome et al., 2019a*; *Lopez-Jacome et al., 2019b*), highlighting the need for therapies that target both QS and T3SS. Previous studies (*Sethupathy et al., 2016*; *Bahari et al., 2017*) showed that curcumin, alone or combined with antibiotics like gentamicin and azithromycin, inhibits (QS)-regulated virulence factors. In this study, we corroborated that curcumin significantly inhibited QS-regulated virulence factors, with the strongest effect observed on pyoverdine production a siderophore that is crucial for *P. aeruginosa* fitness during infections (*Jeong et al., 2023*). Hence, curcumin's dual inhibitory effect on QS-regulated virulence factors and in T3SS makes it a versatile candidate for comprehensive anti-virulence strategies.

The variable efficacy of curcumin across clinical isolates is noteworthy. While curcumin completely inhibited ExoS and ExoU secretion in PA01, PA14, and some clinical isolates (*e.g.*, INP-40, P076, P193), its effect on ExoS secretion in INP-30R and P044 was minimal. This strain-specific response may reflect genetic or phenotypic differences,

such as variations in T3SS regulatory pathways or membrane permeability to curcumin. Additionally, the observation that curcumin slightly increased exoprotease activity in P044 suggests complex interactions with certain virulence pathways, warranting further investigation.

The clinical isolates studied here exhibited a range of antibiotic resistance profiles, from sensitive (INP-40, P193) to extensively drug-resistant (P044). Curcumin's efficacy against MDR and extensively drug-resistant strains is particularly encouraging, as it offers a non-bactericidal approach that minimizes selective pressure for resistance development. Moreover, curcumin's lack of significant growth inhibition at 50 µM confirms its anti-virulence specificity, reducing the risk of disrupting commensal microbiota or promoting resistance.

However, several challenges remain. Curcumin's poor bioavailability and solubility limit its clinical utility, necessitating the development of formulations (*e.g.*, nanoparticles or liposomes) to enhance delivery (*Tabanelli, Brogi & Calderone, 2021*). Nevertheless, we envision that topical application of curcumin to prevent or eliminate *P. aeruginosa* infections in wounds is suitable, since several curcumin topical formulations had been tested in rodents and were able to accelerate wound healing (*Mohanty & Sahoo, 2017*). The dual effects of promoting healing and inhibiting *P. aeruginosa* virulence make curcumin a very attractive compound. Moreover, the excellent safety profile of curcumin makes it possible to administrate high topical concentrations that will surpass the ones used in our *in vitro* study (50 µM) as well as those used in other *in vitro* studies that demonstrated the inhibition of QS-dependent virulence factors in *P. aeruginosa* (from 13.5 to 86.7 µM) (*Sethupathy et al., 2016*; *Bahari et al., 2017*).

Moreover, the strain-specific effects observed here suggest that curcumin alone may not be effective to mitigate virulence of all *P. aeruginosa* strains present in infections. Hence, further exploring curcumin's synergy with existing antibiotics or QS inhibitors could enhance its therapeutic potential.

Since our study was limited due to lack of elucidation of mechanistic details about curcumin mediated T3SS inhibition, future studies should focus on elucidating the precise molecular mechanism by which curcumin disrupts T3SS function, like curcumin's possible interference with needle assembly or effector translocation.

Moreover, *in vivo* studies using animal models of *P. aeruginosa* infection are critical to validate curcumin's efficacy and assess its impact on host survival and immune responses.

## CONCLUSIONS

Curcumin inhibits the secretion of type III effectors of the two main reference *P. aeruginosa* strains PA01 and PA14 and of clinical strains, from cystic fibrosis and burn patients, including multidrug and extensive drug resistant strains. Beyond T3SS inhibition, curcumin also inhibited pyoverdine production in all tested strains, caseinolytic activity in the two reference strains and one clinical isolate and biofilm formation in one clinical strain.

Our study suggests that this natural compound may be useful for the treatment of acute *P. aeruginosa* infections and encourages further research towards the generation of novel therapies against this recalcitrant bacterium.

### Funding

Miguel Diaz-Guerrero was supported by a Dirección General de Asuntos del Personal Académico, UNAM. This work was funded by PAPIIT, UNAM grants: IN200121 and IN200224 to Rodolfo García-Contreras. The funders had no role in study design, data collection and analysis, decision to publish, or preparation of the manuscript.

### Grant Disclosures

The following grant information was disclosed by the authors:
Dirección General de Asuntos del Personal Académico, UNAM.
PAPIIT, UNAM grants: IN200121, IN200224.

### Competing Interests

Rodolfo García-Contreras is an Academic Editor for PeerJ.

### Author Contributions

- Miguel Diaz-Guerrero conceived and designed the experiments, performed the experiments, analyzed the data, prepared figures and/or tables, authored or reviewed drafts of the article, and approved the final draft.
- Luis Esaú López-Jácome conceived and designed the experiments, analyzed the data, authored or reviewed drafts of the article, and approved the final draft.
- Rafael Franco-Cendejas conceived and designed the experiments, analyzed the data, authored or reviewed drafts of the article, and approved the final draft.
- Rafael Coria-Jiménez conceived and designed the experiments, analyzed the data, authored or reviewed drafts of the article, and approved the final draft.
- María Guadalupe Martínez-Zavaleta performed the experiments, authored or reviewed drafts of the article, and approved the final draft.
- Bertha González-Pedrajo conceived and designed the experiments, analyzed the data, authored or reviewed drafts of the article, and approved the final draft.
- Daniel Huelgas-Méndez performed the experiments, analyzed the data, prepared figures and/or tables, authored or reviewed drafts of the article, and approved the final draft.
- Rodolfo García-Contreras conceived and designed the experiments, analyzed the data, prepared figures and/or tables, authored or reviewed drafts of the article, and approved the final draft.

### Human Ethics

The following information was supplied relating to ethical approvals (i.e., approving body and any reference numbers):

The research committee of the National Institute of Rehabilitation in Mexico approved the study and the collection of the clinical strains used.

## Data Availability

The raw data is available in the Supplemental File.

## Supplemental Information

Supplemental information for this article can be found online at http://dx.doi.org/10.7717/peerj.19725#supplemental-information.

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
