# Peer review of "Curcumin inhibits type III secretion of Pseudomonas aeruginosa"

_PeerJ, doi:10.7717/peerj.19725_

## Round 0.1 · original submission · Major Revisions

·

Basic reporting

The manuscript is well-written, and the research question is clearly defined. The background information regarding the importance of the Type III Secretion System (T3SS) in Pseudomonas aeruginosa virulence is well-explained, and the rationale for using curcumin as a potential anti-virulence compound is supported by relevant literature. The results are clearly presented, and the figures and graphs effectively complement the data. The visual representation of results enhances the clarity of the findings, providing easy comprehension of the study’s outcome.

Strengths:
Well-structured introduction with a clear explanation of the research question.
Adequate referencing and context from prior studies.
Visual elements (figures/graphs) are appropriate and help in understanding the results.

Weaknesses:
Minor language polishing could be beneficial to improve clarity and readability in some sections.

Experimental design

The experimental design is robust, with well-chosen strains and detailed methodology for testing curcumin's effects on T3SS-mediated secretion. The manuscript employs both laboratory reference strains (PA14, PA01) and clinical isolates, strengthening the relevance of the findings. The use of positive and negative controls is well-justified, and the description of experimental procedures is thorough and reproducible. The inclusion of visual data (such as Western blot images and biofilm assays) effectively supports the validity of the results.

Strengths:

Clear rationale for strain selection, including clinical isolates and reference strains.

Use of multiple experimental controls to validate findings.

Adequate explanation of methodologies, including the use of graphical data to support conclusions.

Weaknesses:

The authors should provide further discussion on the rationale for the curcumin concentration used (50 µM), particularly in relation to in vivo applicability.

It would be useful to include additional detail about how clinical isolates were selected from the available collection.

Validity of the findings

The results indicate that curcumin inhibits the secretion of ExoS and ExoU effectors in P. aeruginosa strains without significantly affecting bacterial growth, supporting the potential of curcumin as an anti-virulence compound. The Western blot analysis and biofilm assays, which are presented in the figures, substantiate the inhibition of T3SS-mediated secretion. The finding of increased intracellular levels of ExoS/ExoU, despite their inhibition in secretion, is intriguing but requires further exploration.

Strengths:

Solid experimental design and data collection.
Clear evidence that curcumin inhibits ExoU and ExoS secretion without affecting bacterial growth.
Figures and graphical data effectively highlight key findings (e.g., Western blot results).

Weaknesses:
The authors should provide further mechanistic insight into the intracellular accumulation of ExoS and ExoU. This phenomenon is not fully explained in the discussion and warrants deeper exploration.

Additional comments

The authors should polish the language in some sections to improve readability and ensure the scientific tone is consistent throughout.

The observation of intracellular accumulation of ExoU and ExoS suggests an interesting aspect of curcumin’s action that could be explored in future studies.

The manuscript could benefit from a discussion on the broader implications of curcumin’s potential application in vivo and its limitations.

Suggestions for Improvement:

Consider discussing in more detail the physiological relevance of the 50 µM curcumin concentration.

Further investigation into the mechanism by which curcumin inhibits the secretion of effectors while increasing their intracellular accumulation would add value to the manuscript.

The inclusion of more extensive discussion on the potential applications of curcumin in treating infections caused by P. aeruginosa would be beneficial.

Reviewer 2 ·

Basic reporting

Introduction is not very clear. Too lengthy and there is not flow between paragraphs or paragraphs are disconnected. Several unconnected matter in the introduction to be deleted. There is need for thorough revision of introduction making the research hypothesis clear. English language used to be unambiguous, lengthy sentences to be revised in to short meaningful sentences.

Literature references, sufficient field background/context provided.
References are given in results section. Ideally results section of the manuscript should present the results of the current study in a systematic manner as per the objectives of the study. Which is lacking in this manuscript. some of the references are old.

Professional article structure, figures, tables. Raw data shared.
Article structure looks professional but the written matter is not professional.
figures, tables. Raw data are shared

Self-contained with relevant results to hypotheses.
Relevance of results to hypotheses are lacking. Many of the current study results is missing {caseinolytic activity, biofilm, cell associated protein & cell supernatant proteins…}. Authors have given results of earlier works! No clarity in results. Needs to be revised as per the hypothesis & experimental objectives.

Experimental design

Original primary research within Aims and Scope of the journal.
yes
Research question well defined, relevant & meaningful. It is stated how research fills an identified knowledge gap.
Research question to be well defined. Authors need to clearly define how research fills an identified knowledge gap.

Rigorous investigation performed to a high technical & ethical standard.
yes

Methods described with sufficient detail & information to replicate.
Methods needs to be clearly described with sufficient information to replicate.
There is no clarity on which strains & their exact source
Strains used , their origin, the media used & growth conditions can be given in a tabular form.
MIC and antibiotic tested not mentioned in material & methods. In what way this is connected to T3SS needs to be justified
Details on kinetic curve is lacking
Authors state LB or LB modified medium is used. Whether both are used or only one type is used? Needs justification
Source & purity of curcumin to be mentioned
why only pyoverdine ? why not pyocyanin or pyorubin or other pigments? needs justification
Biofilm: round or flat bottom places used? how was the 96 well plates sterilized?
What do the authors mean by spend media?
How many times the biofilm assay was repeated? how many wells were used for each isolate in an experiment needs to be mentioned
Type III mediated secretion assay : needs more clarity on how the cell associated & cell supernatant proteins were extracted? how these proteins are related to Type III secretion system?
source of antibodies not mentioned

Validity of the findings

Impact and novelty not assessed. Meaningful replication encouraged where rationale & benefit to literature is clearly stated.
-Impact of the study not discussed. Needs to compare with recent findings.

All underlying data have been provided; they are robust, statistically sound, & controlled.

Data are provided. Cannot comment on the statistics. Pls consult a statistician.
Conclusions are well stated, linked to original research question & limited to supporting results.
Conclusions are not clear & somewhat linked to original research. Information on biofilm caseinolytic activity & pigments are missing.
Limitations of the study is missing.

The conclusions should be appropriately stated, should be connected to the original question investigated, and should be limited to those supported by the results. In particular, claims of a causative relationship should be supported by a well-controlled experimental intervention. Correlation is not causation.
Pls refer the comments given in the Pdf of the manuscript

Additional comments

The English language and grammar should be improved to ensure that an international audience can clearly understand the text. Bacterial names to be in Italics.
Some of the references are very old needs to be replaced. Reference Nos.: 10, 22, 24, 28
Figure 1 & supplementary file figure 1 looks alike. Duplication may be avoided
Figure 2 legends need more clarity

Looks like the authors needs to revise the entire manuscript thoroughly, so as to have a continuous flow & relations between introduction, material & methods results, discussion and conclusion.

Annotated reviews are not available for download in order to protect the identity of reviewers who chose to remain anonymous.

---

## Round 0.2 · Minor Revisions

Please consider the recommendations made by Reviewer 2 to polish the text of your manuscript and get it into the best possible shape before publication. I am looking forward to seeing these minor revisions.

·

Basic reporting

The authors have edited the article according to my previous suggestions.

Experimental design

The authors have edited the article according to my previous suggestions.

Validity of the findings

The authors have edited the article according to my previous suggestions.

Reviewer 2 ·

Basic reporting

Authors have incorporated the suggested changes & now it is clear.

Experimental design

Research questions are now well defined. The authors have incorporated most of the required changes

Validity of the findings

Figures are valid

Additional comments

Minor corrections/suggestions are included in the annotated PDF file. Authors may pay attention to English grammar & spelling.

Annotated reviews are not available for download in order to protect the identity of reviewers who chose to remain anonymous.

---

## Round 0.3 · accepted · Accept

The authors have now addressed all of the reviewers' concerns. The latest version of the manuscripts shows the changes that have been made during the last round of revisions. In its present state, the manuscript is ready for publication.